# The ancient CYP716 family is a major contributor to the diversification of eudicot triterpenoid biosynthesis

Karel Miettinen[1,2], Jacob Pollier[1,2], Dieter Buyst[3], Philipp Arendt[1,2,4,5], René Csuk[6], Sven Sommerwerk[6], Tessa Moses[1,2], Jan Mertens[1,2], Prashant D. Sonawane[7], Laurens Pauwels[1,2], Asaph Aharoni[7], José Martins[3], David R. Nelson[8] & Alain Goossens[1,2]

Triterpenoids are widespread bioactive plant defence compounds with potential use as pharmaceuticals, pesticides and other high-value products. Enzymes belonging to the cytochrome P450 family have an essential role in creating the immense structural diversity of triterpenoids across the plant kingdom. However, for many triterpenoid oxidation reactions, the corresponding enzyme remains unknown. Here we characterize CYP716 enzymes from different medicinal plant species by heterologous expression in engineered yeasts and report ten hitherto unreported triterpenoid oxidation activities, including a cyclization reaction, leading to a triterpenoid lactone. Kingdom-wide phylogenetic analysis of over 400 CYP716s from over 200 plant species reveals details of their evolution and suggests that in eudicots the CYP716s evolved specifically towards triterpenoid biosynthesis. Our findings underscore the great potential of CYP716s as a source for generating triterpenoid structural diversity and expand the toolbox available for synthetic biology programmes for sustainable production of bioactive plant triterpenoids.

[1] Department of Plant Systems Biology, VIB, Technologiepark 927, B-9052 Ghent, Belgium. [2] Department of Plant Biotechnology and Bioinformatics, Ghent University, B-9052 Ghent, Belgium. [3] Department of Organic Chemistry, Ghent University, B-9000 Ghent, Belgium. [4] Laboratory for Protein Biochemistry and Biomolecular Engineering, Department of Biochemistry and Microbiology, Ghent University, B-9000 Ghent, Belgium. [5] VIB Medical Biotechnology Center, B-9000 Ghent, Belgium. [6] Department of Organic Chemistry, Martin-Luther-University Halle-Wittenberg, D-06120 Halle (Saale), Germany. [7] Department of Plant and Environmental Sciences, Weizmann Institute of Science, Rehovot 76100, Israel. [8] Department of Microbiology, Immunology and Biochemistry, University of Tennessee Health Science Center, Memphis, Tennessee 38163, USA. Correspondence and requests for materials should be addressed to A.G. (email: alain.goossens@psb.vib-ugent.be).

Triterpenoids and triterpene saponins compose a widespread family of naturally occurring plant defence compounds with an immense diversity in structure and function. They have numerous applications as pharmaceuticals, cosmetics, food and agronomic agents[1,2]. Triterpene saponins are amphipathic molecules that consist of a hydrophobic triterpenoid backbone decorated by sugar chains and other functional groups. The structural diversity of saponins arises from their modular biosynthesis in which three main steps can be distinguished: (1) cyclization of the common precursor 2,3-oxidosqualene by oxidosqualene cyclases (OSCs) generates the different triterpenoid backbones, (2) oxidative decoration of the backbone by cytochrome P450 monooxygenase enzymes (P450s) and (3) addition of sugar (chains) by UDP-glycosyltransferases (UGTs) or other groups to positions activated by the P450s[1–5].

The triterpenoid backbones or sapogenins are known to be oxidized on most of the (primary and secondary) carbon atoms of the 30-carbon backbone and exist in numerous different combinations of oxidized functional groups[1–5]. Although several triterpenoid-decorating P450s from different P450 families representing five (CYP51, 71, 72, 85 and 86) P450 clans (Supplementary Table 1)[3,6–13] have been identified, these enzymes remain relatively unexplored in the plant kingdom. Recruitment of P450s to triterpenoid biosynthesis has been suggested to have occurred several times during evolution in different plant lineages[14,15]. Although some P450 families, such as the triterpenoid-metabolizing enzymes belonging to the CYP72 family (CYP72A subfamily) from legumes seem to be plant specific, others are suggested to be more common, such as the ancient CYP716 enzymes that already emerged in Bryophyta (mosses)[14–16]. CYP716 enzymes are indeed ubiquitous and have been found across most land plants[14]. The CYP716 family belongs to the CYP85 clan of P450s, corresponding to non-A type P450s usually seen as housekeeping enzymes[14,16], which underlines the evolutionary age of this clade and its proposed origin in triterpenoid primary metabolism[14]. By far the most common modification found in triterpene saponins is the three-step oxidation at the C-28 position of β-amyrin, α-amyrin and lupeol, leading to oleanolic acid (oleanane-type), ursolic acid (ursane-type) and betulinic acid (lupane-type) backbones, respectively, which can be found in more than 1,600 plant species from over 140 families[16]. The highly conserved CYP716A subfamily enzymes, exemplified by CYP716A12 from the model legume Medicago truncatula, are the only known enzymes performing this three-step oxidation, and, conversely, most of the characterized CYP716s catalyse C-28 oxidation. Nonetheless, some CYP716s have lately been found to catalyse other triterpenoid oxidation reactions in plant species from different families (across higher eudicots), namely C-16α oxidation of β-amyrin, C-22α oxidation of α-amyrin, C-3 oxidation of β-amyrin, α-amyrin and lupeol, C-12 oxidation of dammarenediol-II, C-6 oxidation of protopanaxadiol and an unknown hydroxylation of tirucalla-7,24-dien-3β-ol[8,9,11,12,17–20]. The occurrence of several CYP716 sequence homologues from different subfamilies in the sequenced land plants (http://drnelson.uthsc.edu/cytochromeP450.html) suggests a great but yet unexplored potential for discovery of new triterpenoid-metabolizing enzymes and marks this family as an interesting subject for phylogenetic and structure-activity studies.

The characterized CYP716s listed above were largely discovered following targeted transcriptome analyses or mutant screens in species from different orders in the Rosid and Asterid clades, thereby spanning the core eudicot clade (according to the APG III System[21]). Given the recurrent identification of CYP716s in the above-listed 'triterpenoid-oriented' screens, we decided to launch a 'CYP716-oriented' screen in medicinal plant species spanning the eudicot clade that produce interesting triterpenoids with decorations generated by yet unknown enzymes. The selected plants were Centella asiatica and Platycodon grandiflorus from the Apiales and Asterales orders (both Asterids), respectively, and the basal eudicot Aquilegia coerulea from the Ranunculales order. Together, 30 CYP716s were identified in the transcriptomes and genomes of these species of which 10 were found to catalyse a variety of different triterpenoid oxidation reactions, many of which were previously unknown, such as a cyclization reaction leading to a triterpenoid 13,28-lactone. The fact that at least one third of the analysed CYP716 enzymes showed triterpenoid-oxidizing capacity emphasizes the triterpenoid-oriented activity of the CYP716 family and its important role in triterpenoid biosynthesis in eudicots. This is supported by an extensive phylogenetic analysis, including the output of the 1,000 plants (1KP) initiative (https://sites.google.com/a/ualberta.ca/onekp/), which confirms the existence of eudicot-type CYP716s and indicates they have been lost in monocots.

## Results

**CYP716 enzymes in C. asiatica triterpenoid biosynthesis.** The medicinal plant C. asiatica produces a variety of triterpene saponins, including asiaticoside and madecassoside, two ursane-type triterpene saponins widely studied for their pharmaceutical properties[22,23] (Fig. 1a). Up to now, the biosynthesis of these compounds remains largely unknown and is believed to include four oxidative modifications catalysed by P450s. Considering the reported implication of CYP716s in the oxidation of triterpenoid backbones at various positions[11,12,17–19], we mined publicly available C. asiatica transcriptome data[24] for CYP716-encoding genes. This analysis revealed six candidate CYP716 enzymes (Supplementary Table 2a) with 46–78% amino acid identity with known CYP716 proteins. Five of the six corresponding genes could be cloned from a custom-made complementary DNA (cDNA) library and the obtained sequences were submitted (as for all CYP716s from this study) to the P450-naming committee and GenBank under the acronyms CYP716A86, CYP716A83, CYP716D36, CYP716E41 and CYP716C11, and accession numbers KU878848–KU878852, respectively. All five P450s were heterologously expressed in a Saccharomyces cerevisiae strain engineered for triterpenoid production[19] together with a β-amyrin synthase from Glycyrrhiza glabra (GgBAS)[19,25] and a cytochrome P450 reductase from M. truncatula (MTR1; Medtr3g100160; GenBank accession KU878869). Gas chromatography–mass spectrometry (GC–MS) analysis of pooled hexane and ethyl acetate extracts of the spent medium of methyl-β-cyclodextrin (MβCD)-treated yeast cultures revealed three new peaks for yeast expressing CYP716A83 and CYP716A86 (Fig. 1b). These peaks correspond to erythrodiol, oleanolic aldehyde and oleanolic acid (Supplementary Fig. 1a), implicating that, similarly to CYP716A12 from M. truncatula[26], CYP716A83 and CYP716A86 catalyse the three-step oxidation of β-amyrin at the C-28 position to yield oleanolic acid. In our engineered yeast background, CYP716A83 appeared to be functionally more efficient than CYP716A86.

To further investigate their potential role in C. asiatica triterpene saponin biosynthesis, the remaining three CYP716 enzymes were expressed in our yeast together with CYP716A83, GgBAS and MTR1. GC–MS analysis of the spent medium of the MβCD-treated yeast cultures revealed that CYP716C11 and CYP716E41 converted oleanolic acid to maslinic acid (2α-hydroxy oleanolic acid) and two unknown compounds, respectively (Fig. 1c). The unknown compounds have a molecular ion at m/z 688 Da and m/z 616 Da, and probably correspond to

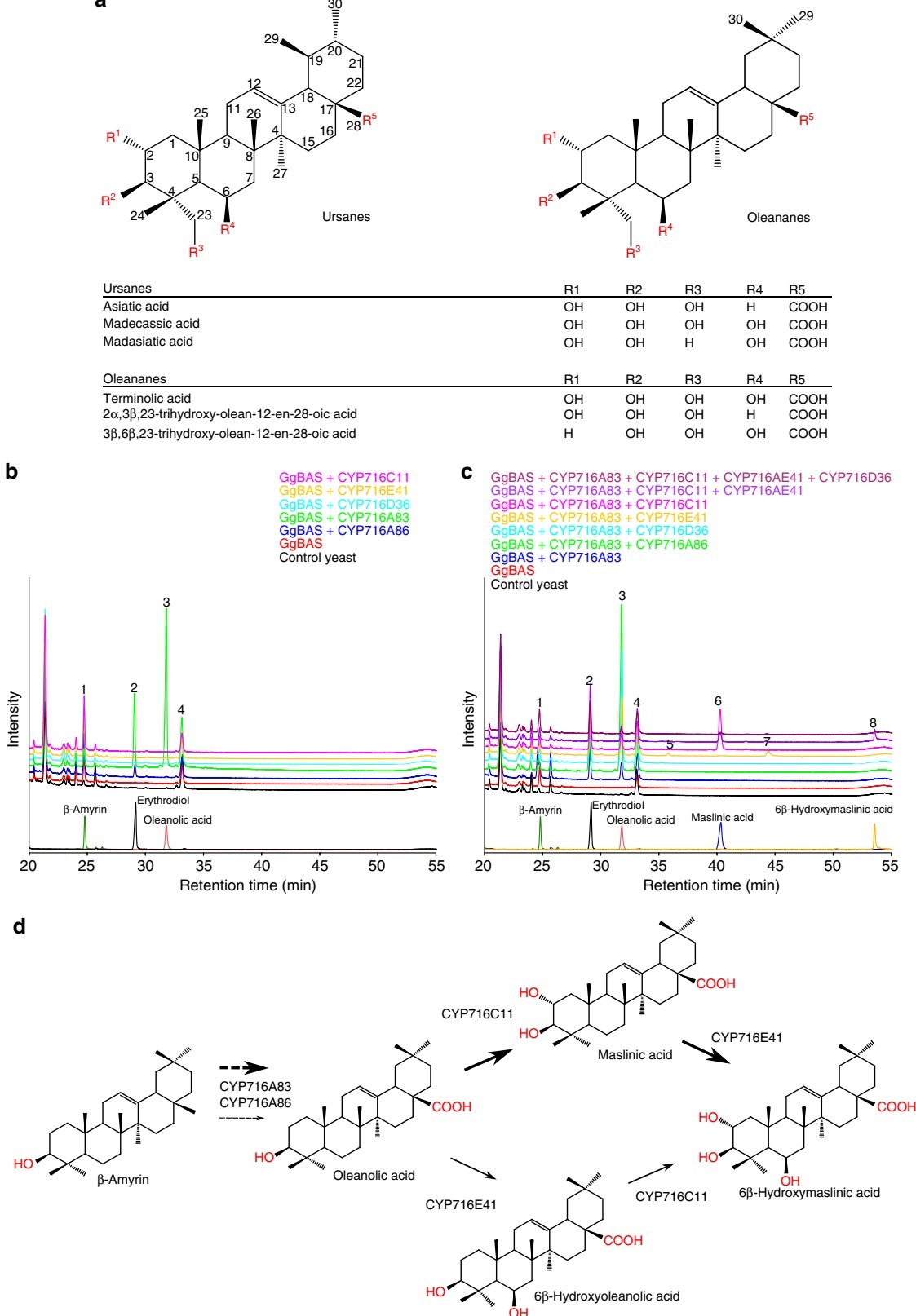

**Figure 1 | *C. asiatica* sapogenins and *in vivo* functional analysis of *C. asiatica* CYP716s.** (**a**) Examples of oleanane- and ursane-type sapogenins found in *C. asiatica*. (**b**) Overlay of GC–MS total ion current chromatograms showing accumulation of standard compounds and triterpenoids produced in yeast strains expressing single *C. asiatica CYP716*s. (**c**) Overlay of GC–MS total ion current chromatograms showing accumulation of standard compounds and triterpenoids produced in yeast strains expressing *C. asiatica CYP716*s in combination with *CYP716A83*. Annotated triterpenoid peaks are indicated with numbers: (1) β-amyrin, (2) erythrodiol, (3) oleanolic acid, (4) oleanolic aldehyde (which co-elutes with a nonspecific peak), (5) 6β-hydroxy β-amyrin, (6) maslinic acid, (7) putative incompletely derivatized 6β-hydroxy oleanolic acid and (8) 6β-hydroxy maslinic acid. (**d**) Proposed biosynthetic pathway of sapogenins in *C. asiatica*. Arrows of higher weight reflect the preferred order of reactions in the biosynthetic pathway, as supported by the semi-quantitative analysis shown in Table 1.

**Table 1 | Semi-quantitative analysis of CYP716 substrates and products in transformed yeasts.**

| Compounds produced by *C. asiatica* CYP716s | % β-Amyrin [218]+ | % Oleanolic acid [203]+ | % 6β-OH oleanolic acid [203]+ | % Maslinic acid [203]+ |
|---|---|---|---|---|
| Yeast strain | Compared with strain 2 | Compared with strain 4 | Compared with strain 10 | Compared with strain 11 |
| 2/ BAS | 100 ± 12 | | | |
| 3/ BAS + CYP716A86 | 104 ± 9 | | | |
| 4/ BAS + CYP716A83 | 28 ± 3 | 100 ± 2 | | |
| 10/ BAS + CYP716A83 + CYP716E41 | 36 ± 2 | 25 ± 4 | 100 ± 56 | |
| 11/ BAS + CYP716A83 + CYP716C11 | 35 ± 10 | 10 ± 1 | | 100 ± 11 |
| 14/ BAS + CYP716A83 + CYP716E41 + CYP716C11 | 25 ± 2 | 6 ± 1 | 0 ± 0 | 5 ± 1 |

| Compounds produced by *P. grandiflorus* CYP716s | % β-Amyrin [218]+ | % Oleanolic acid [203]+ | % 16β-OH β-amyrin [216]+ | % 12,13α-epoxy-β-amyrin [514]+ | % 16β-OH oleanolic acid [318]+ |
|---|---|---|---|---|---|
| Yeast strain | Compared with strain 2 | Compared with strain 30 | Compared with strain 32 | Compared with strain 33 | Compared with strain 36 |
| 2/ BAS | 100 ± 12 | | | | |
| 30/ BAS + CYP716A140 | 11 ± 1 | 100 ± 4 | | | |
| 32/ BAS + CYP716A141 | 26 ± 2 | | 100 ± 4 | | 29 ± 7 |
| 33/ BAS + CYP716AS5 | 114 ± 12 | | | 100 ± 11 | |
| 36/ BAS + CYP716A140 + CYP716A141 | 7 ± 1 | 24 ± 3 | 10 ± 0 | | 100 ± 25 |
| 37/ BAS + CYP716A140 + CYP716S5 | 18 ± 1 | 117 ± 7 | | 0 ± 0 | |
| 39/ BAS + CYP716A140 + CYP716A141 + CYP716S5 | 7 ± 1 | 30 ± 3 | 10 ± 0 | 0 ± 0 | 103 ± 26 |

| Compounds produced by *A. coerulea* CYP716s | % β-Amyrin [218]+ |
|---|---|
| Yeast strain | Compared with strain 2 |
| 2/ BAS | 100 ± 12 |
| 46/ BAS + CYP716A110 | 67 ± 8 |
| 47/ BAS + CYP716A111 | 115 ± 8 |

Known triterpenoids were analysed and quantified from spent medium of MβCD-treated yeast cultures. The values indicate % of substrate left in comparison with the strain indicated ± s.e. (n = 4). These percentages are calculated from the means of peak areas of extracted ion intensities of representative ions of known triterpenoids, which are provided in Supplementary Table 3. Production rates of all products were consistent between the four replicates, except for the strains with *C. asiatica* CYP716E41, which showed larger quantitative variability although the same products were present in all four replicates. The comparison of substrate accumulation between producer yeast strains with and without additional CYP716s from *C. asiatica*, *P. grandiflorus* and *A. coerulea* showed diminished accumulation of the substrate compounds with active CYP716s.

(silylated) hydroxy oleanolic acid and incompletely derivatized hydroxy oleanolic acid, respectively. Further combining of CYP716A83, CYP716C11 and CYP716E41 resulted in the production of a new compound with a molecular ion at *m/z* 704 Da. This compound was purified from the spent medium of a MβCD-treated yeast culture and identified by nuclear magnetic resonance (NMR) as 6β-hydroxy maslinic acid (Supplementary Fig. 1a and Supplementary Methods). 6β-Hydroxylation is observed in many *C. asiatica* saponins[23]. This also implicates that the unknown compounds produced by CYP716E41 in the *GgBAS/CYP716A83*-expressing yeast probably correspond to 6β-hydroxy oleanolic acid and incompletely derivatized 6β-hydroxy oleanolic acid (Fig. 1b–d). Expression of *CYP716D36*, either alone or in combination with one or more of the CYP716A83, CYP716E41 or CYP716C11 enzymes did not lead to the production of new metabolites (Fig. 1b,c).

Expression of the pairs *CYP716A83/CYP716E41* and *CYP716A83/CYP716C11* in a yeast strain expressing the *C. asiatica* dammarenediol synthase (CaDDS), which was shown to produce α-amyrin, β-amyrin and minute amounts of dammarenediol-II[19], led to the production of the same oleanane products as with GgBAS, as well as of the corresponding ursane products, namely putative uvaol, ursolic acid and putative 6β-hydroxy ursolic acid for CYP716A83/CYP716E41, and putative uvaol, ursolic acid and corosolic acid for CYP716A83/CYP716C11 (Supplementary Fig. 2). Ursolic acid and corosolic acid were confirmed with authentic standards (Supplementary Fig. 1b). GC–MS analysis of a yeast strain expressing the combination CYP716A83/ CYP716E41/CYP716C11 did not reveal additional products, indicating that neither 6β-hydroxy maslinic acid nor

the corresponding ursane 6β-hydroxy corosolic acid was produced in this background, which may be due to the overall low levels of α-amyrin and β-amyrin precursors (Supplementary Fig. 2). Our results implicate that CYP716E41 and CYP716C11 are specific for C-28 carboxylated triterpenoids, and that CYP716A83, CYP716C11 and CYP716E41 can function on both oleanane and ursane backbones, which also reflects the presence of oleanane- and ursane-type triterpene saponins in the plant. Although CaDDS can also generate dammarenediol-II[19], and dammarane-type saponins are found in *C. asiatica*[27], no oxidized dammarane-type compounds were observed in yeast expressing *CaDDS* together with any combination of the *C. asiatica* CYP716s. However, as only minute amounts of the dammarenediol-II precursor are produced in our yeast strain, we cannot entirely rule out that this triterpenoid backbone could still be a substrate for the *C. asiatica* CYP716s characterized here.

Taken together, characterization of the *C. asiatica* CYP716 family reveals members with three different activities, two of which have previously unreported specificity for the 2α- and 6β-position on the pentacyclic triterpenoid skeleton. Finally, semi-quantitative analysis of the identified triterpenoids was performed to determine the substrate consumption upon introduction of one or more CYP716 enzymes and thereby define substrate preferences of the *C. asiatica* CYP716 enzymes (Table 1 and Supplementary Table 3). Based on this analysis, we propose the pathway model shown in Fig. 1d.

**Functionally versatile CYP716s from *P. grandiflorus*.** Sapogenins of most common saponins contain functional groups that are the product of simple oxidation reactions, leading to hydroxyl,

carbonyl or carboxyl groups. The success of our approach with the *C. asiatica* enzymes raised the hypothesis that CYP716 enzymes could also be responsible for less common functional groups. *P. grandiflorus*, a traditional medicinal plant, accumulates saponins in its roots, including several with 13,28β- and

2,24β-lactone rings (Fig. 2a)[28]. No enzymes catalysng the biosynthesis of such structures on triterpenoid backbones are known to date. Sequence resources for this plant were obtained from 'The Compositae Genome Project' (http://compgenomics.ucdavis.edu/) and mined using the BLASTX

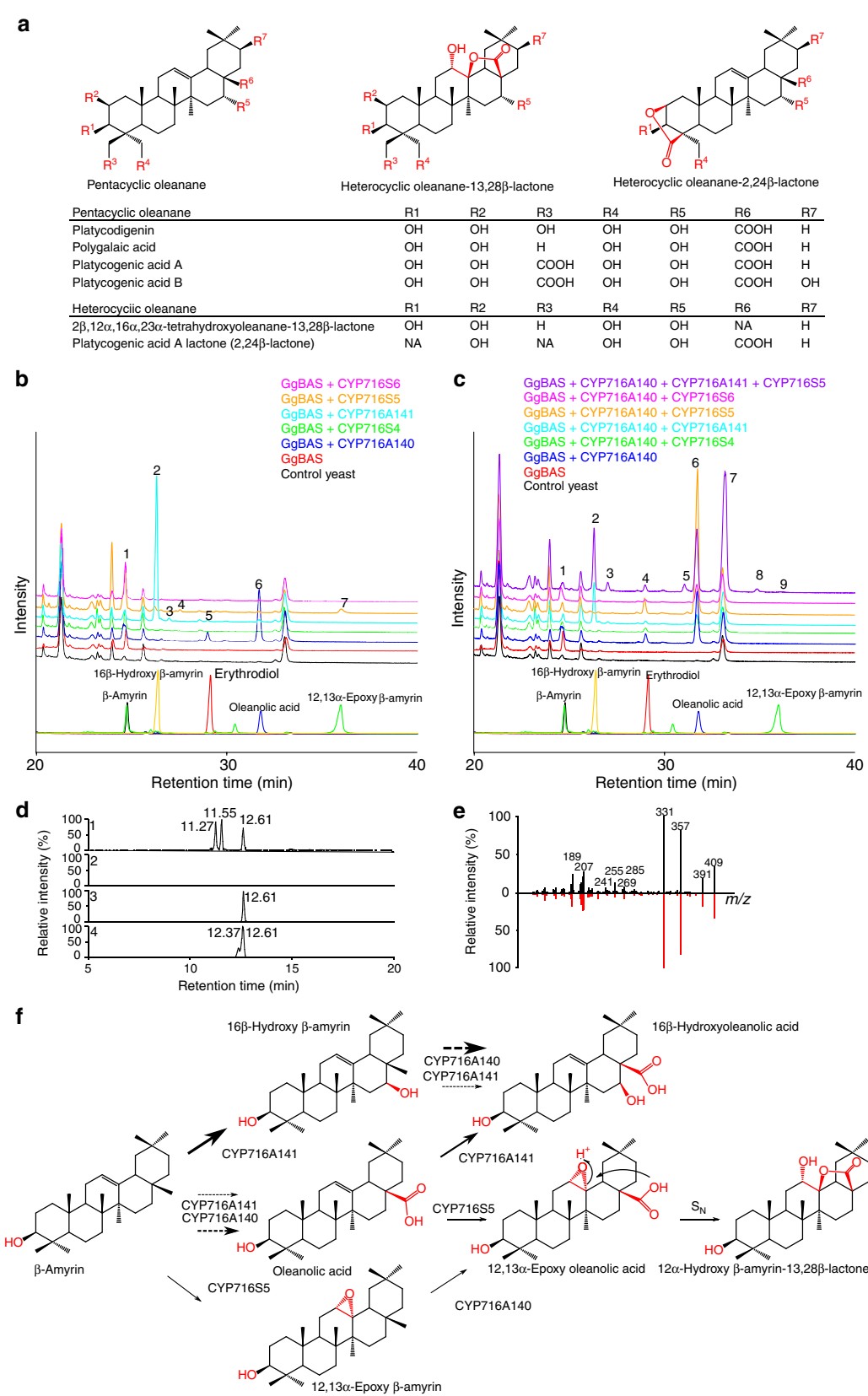

algorithm, yielding six CYP716s in the available databases (Supplementary Table 2b). Five of the candidate sequences were incomplete; hence, rapid amplification of cDNA ends (RACE) PCR on *P. grandiflorus* seedling cDNA was carried out to obtain full-length open reading frame (FL-ORF) sequences. FL-ORF sequences were obtained for five of the six identified CYP716s, which were named CYP716A140, CYP716A141, CYP716S4, CYP716S5 and CYP716S6 (GenBank accession numbers KU878853–KU878857).

The *P. grandiflorus* CYP716s were heterologously expressed in yeast together with *GgBAS* and *MTR1*, and analysed as described above. GC–MS analysis of the spent medium of the MβCD-treated cultures of CYP716A140-expressing yeast showed three new peaks corresponding to C-28 oxidation (Fig. 2b). The expression of CYP716S5 in this background produced two new peaks (Fig. 2b); one with a parent mass of *m/z* 514 Da and another smaller peak with a parent mass of *m/z* 586 Da. The former compound was identified as 12,13α-epoxy β-amyrin using an authentic standard (Supplementary Fig. 1c), but the identity of the latter compound remains unknown. Its mass suggests a (silylated) hydroxy β-amyrin. Expression of CYP716A141 led to a new peak (Fig. 2b) with a molecular ion at *m/z* 586 Da, a mass corresponding to (silylated) hydroxy β-amyrin. This compound was purified from the spent medium of a MβCD-treated yeast culture and analysed by NMR, which established its structure unambiguously as that of 16β-hydroxy β-amyrin (Supplementary Fig. 1c and Supplementary Methods). CYP716A141 additionally produced some minor peaks among which two had the same retention time and parent masses as the erythrodiol and oleanolic acid standards (Fig. 2b). Hence, CYP716A141 appears to be a multifunctional β-amyrin oxidase.

To assess whether any of the *P. grandiflorus* CYP716s could use any of the CYP716A140 products as a substrate, they were all analysed in a yeast strain expressing *GgBAS* and CYP716A140. Only one of the combinations tested led to the production of a new compound; CYP716A141 expressed together with CYP716A140 resulted in a new peak with a parent mass at *m/z* 688 Da, probably corresponding to 16β-hydroxy oleanolic acid (Fig. 2c). Finally, expression of a third P450, that is CYP716S5 in the strain already expressing CYP716A141 and CYP716A140 with *GgBAS* led to two additional new peaks with the apparent molecular ion at *m/z* 674 and 600 Da, which possibly represent a β-amyrin di-alcohol and a β-amyrin with an additional aldehyde and hydroxyl group, respectively (Fig. 2c). Based on the demonstrated activities of CYP716A140 and CYP716S5, expressing both of them in combination with *GgBAS* in yeast was expected to produce 12,13α-epoxy-oleanolic acid (expected parent mass *m/z* 616 Da)[29] or hydroxy oleanolic acid (expected parent mass *m/z* 688 Da). However, no new peaks corresponding to these compounds were observed (Fig. 2c). As *P. grandiflorus* is also known to accumulate heterocyclic saponins with a 13,28β-lactone structure[30], we therefore hypothesized that such a bridge structure could be the product of the combined activities of CYP716A140 and CYP716S5, which would yield a 12α-hydroxy β-amyrin-13,28β-lactone. Yet, no peaks corresponding to such a compound (expected parent mass *m/z* 616 Da) could be detected either (Fig. 2c), possibly because it may be thermolabile and therefore not traceable by GC–MS analysis. To analyse *P. grandiflorus* CYP716 products with a less intrusive method, the ethyl acetate extract of the spent medium of MβCD-treated yeast culture co-expressing *GgBAS*, CYP716A140 and CYP716S5 was analysed with liquid chromatography–atmospheric pressure chemical ionization–Fourier transform–ion cyclotron resonance MS (LC-APCI-FT-ICR-MS) (Fig. 2d). This revealed three new peaks with an exact mass of 473.3622 Da, with one of them corresponding to the heterocyclic 12α-hydroxy β-amyrin-13,28βlactone standard (Fig. 2e). To assess whether one of the other unknown peaks corresponded to 12,13α-epoxy oleanolic acid, the anticipated direct product of the combination of CYP716A140 and CYP716S5, a standard of this compound was acquired and also analysed by LC-APCI-FT-ICR-MS (Fig. 2d). This standard showed two peaks with, as expected, the same parent mass (473.3622 Da). One of them perfectly matched 12α-hydroxy β-amyrin-13,28β-lactone and the other was deduced to be 12,13α-epoxy oleanolic acid. Based on this, we postulate that 12,13α-epoxy oleanolic acid is (spontaneously) converted into 12α-hydroxy β-amyrin-13,28β-lactone (Fig. 2f). Accordingly, none of the peaks from the analysis of the yeast expressing CYP716A140 and CYP716S5 corresponded to 12,13α-epoxy oleanolic acid.

The cyclization leading to 12α-hydroxy-β-amyrin-13,28β-lactone is known to happen by a non-enzymatic reaction in some conditions[29]. In a plausible reaction mechanism, the acid-catalysed nucleophilic substitution of epoxides with weak nucleophiles (such as carboxylic acids) would lead to a hydroxy group (here 12α-hydroxy) on the least substituted carbon of the epoxide such as in the 12α-hydroxy β-amyrin-13,28β-lactone. In the nucleophilic substitution reaction ($S_N1$-like), the epoxy group is first protonated, then followed by a back-side nucleophilic attack by the C-28 carboxy group on the weakened C-13-O bond of the epoxide and ended by electron transfer leading to the C-12α hydroxy group (Fig. 2f).

Taken together, analysis of five CYP716s from *P. grandiflorus* yielded three triterpenoid oxidation activities including hitherto unreported and less common reactions such as epoxidation and the subsequent cyclization reaction leading to a triterpenoid lactone. Semi-quantitative analysis, carried out as for the *C. asiatica* CYP716s, allowed determining the preferred substrates of the *P. grandiflorus* CYP716s and inferring the most probable order of catalysis in the pathway (Table 1, Supplementary Table 3 and Fig. 2f).

**Figure 2 | *P. grandiflorus* sapogenins and *in vivo* functional analysis of *P. grandiflorus* CYP716s.** (**a**) Examples of *P. grandiflorus* sapogenins. (**b**) Overlay of GC–MS total ion current chromatograms (TICs) showing accumulation of standard compounds and triterpenoids produced in yeast strains expressing single *P. grandiflorus* CYP716s. Annotated triterpenoid peaks are indicated with numbers: (1) β-amyrin, (2) 16β-hydroxy β-amyrin, (3) putative hydroxy β-amyrin, (4) putative hydroxy β-amyrin, (5) erythrodiol, (6) oleanolic acid and (7) 12,13α-epoxy β-amyrin. (**c**) Overlay of GC–MS TIC showing accumulation of standard compounds and triterpenoids produced in yeast strains expressing *P. grandiflorus* CYP716s in combination with CYP716A140. Annotated triterpenoid peaks are indicated with numbers: (1) β-amyrin, (2) 16β-hydroxy β-amyrin, (3) putative hydroxy β-amyrin, (4) erythrodiol, (5) unknown triterpenoid, (6) oleanolic acid, (7) putative 16β-hydroxy oleanolic acid (co-elutes with an impurity in common for all samples), (8) unknown triterpenoid and (9) 12,13α-epoxy β-amyrin. (**d**) LC-APCI-FT-ICR-MS analysis of the yeast strains expressing *GgBAS* with CYP716A140 and CYP716S5 (1) and *GgBAS* with CYP716A140 only (2), and authentic standards of 12α-hydroxy β-amyrin-13,28β-lactone (3) and 12,13α-epoxy oleanolic acid (4). Chromatograms represent a mass range of *m/z* 473.3622–473.3625 Da. (**e**) Comparison of the MS[3] (473.36→427) fragmentation of the peak at retention time 12.61 min compared with the MS[3] fragmentation of an authentic 12α-hydroxy β-amyrin-13,28β-lactone standard. (**f**) The proposed biosynthetic pathway of triterpene saponins in *P. grandiflorus*. Arrows of higher weight reflect the preferred order of reactions in the biosynthetic pathway, as supported by the semi-quantitative analysis shown in Table 1.

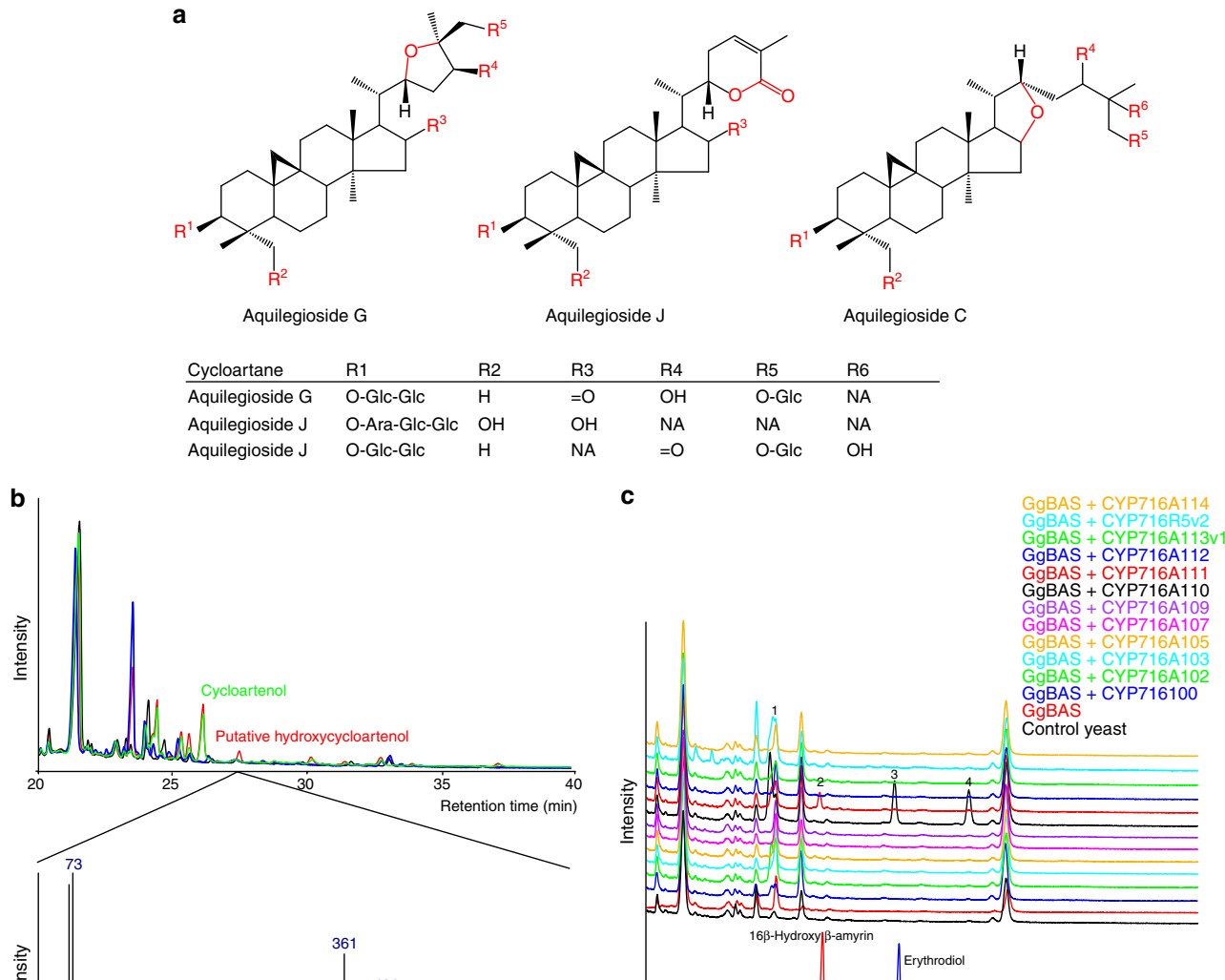

**Figure 3 | Sapogenins from the *Aquilegia* genus and new triterpenoid compounds produced by *A. coerulea* CYP716s. (a)** Examples of sapogenins from *A. vulgaris* including three heterocyclic cycloartanes. **(b)** Overlay of GC–MS total ion current chromatograms showing accumulation of triterpenoids produced in yeast strains expressing *SlCAS* together with *CYP716A113* (red), *SlCAS* only (green), or *CYP716A113* only (blue), compared with a control (empty) strain (black). **(c)** Overlay of GC–MS total ion current chromatograms showing accumulation of standard compounds and triterpenoids produced in yeast strains expressing single *A. coerulea* CYP716s in combination with *GgBAS*. Candidate-specific peaks are indicated with numbers (1) β-amyrin, (2) 16β-hydroxy β-amyrin, (3) erythrodiol and (4) oleanolic acid.

**A. coerulea CYP716s oxidize cycloartanes and oleananes**. The eudicots *C. asiatica* and *P. grandiflorus* belong to the subclade Campanulidae within the Asterids (in the APG III System[21]) and are thus fairly closely related. To expand the taxonomic range of this study, the Colorado blue columbine, *A. coerulea*, was incorporated in this study. This species is considered a model to study the evolutionary separation between eudicots and monocots, because it belongs to the Ranunculaceae family, corresponding to the clade of basal eudicots (*A. coerulea* Genome Sequencing Project http://phytozome.jgi.doe.gov/). Moreover, and of interest to our study, *Aquilegia vulgaris*, a close relative of *A. coerulea*, is known to accumulate cycloartane saponins oxidized at the C-16, C-22, C-25 and C-26 positions, and contains uncommon structures such as three different ether and ester (lactone) rings[31–34] (Fig. 3a).

No enzymes oxidizing cycloartane triterpenoids are known yet, but a simple BLAST search of the *A. coerulea* genome revealed a stunning 18 hits for the CYP716 enzyme family (Supplementary Table 2c). The corresponding FL-ORF sequences could be identified from a gene model prediction data set produced by the FGENESH+ programme (*A. coerulea* Genome Sequencing Project http://phytozome.jgi.doe.gov/). Because of the very high similarity between some of the *CYP716* sequences, we only considered 12 of them for further functional analysis, which were all successfully cloned from cDNA or genomic DNA (gDNA) prepared from *A. coerulea* seedlings and which were named *CYP716A100*, *CYP716A102*, *CYP716A103*, *CYP716A105*, *CYP716A107*, *CYP716A109*, *CYP716A110*, *CYP716A111*, *CYP716A112*, *CYP716A113v1*, *CYP716A114* and *CYP716A114R5v2* (GenBank accession numbers KU878858–KU878868 and KY04760000).

The *A. coerulea* CYP716s were heterologously expressed in yeast engineered for triterpenoid production[19], first in combination with cycloartenol synthase from tomato (*Solanum lycopersicum*) (*SlCAS*; Solyc04g070980; GenBank accession number NM_001246855) and *MTR1*. GC–MS analysis of the pooled hexane and ethyl acetate extract of the spent medium from the MβCD-treated cultures showed several new peaks only for the yeast expressing *CYP716A113v1* (Supplementary Fig. 3a). Although one of the peaks showed a molecular ion at *m/z* 586 Da corresponding to putative (silylated) hydroxy-cycloartenol, others had lower than expected parent masses, raising the possibility that the new compounds resulted from CYP716A113v1 oxidizing endogenous yeast metabolites rather than cycloartenol. To test this hypothesis, all the candidates were expressed in a yeast without any heterologous plant OSCs, which indeed again resulted in new GC–MS peaks for the yeast expressing *CYP716A113v1* (Supplementary Fig. 3b). Comparative analysis of the yeasts expressing either *CYP716A113v1*, *SlCAS* or both, indicated that only the above-mentioned peak with a parent mass of *m/z* 586 Da was dependent on both SlCAS and CYP716A113v1 (Fig. 3b), thus representing the only peak corresponding to hydroxylated cycloartenol. Comparing the yeast exclusively expressing *SlCAS* with a yeast expressing no plant *OSCs* also revealed several new peaks in addition to cycloartenol (Fig. 3b and Supplementary Fig. 3a), showing that cycloartenol can be metabolized by endogenous yeast enzymes as previously described[35].

Even though oleanane-type saponins have yet to be found from *Aquilegia ssp.*, we suspected that *A. coerulea* CYP716 enzymes could also metabolize oleananes because of high sequence similarity with CYP716A subfamily members, such as CYP716A12 (ref. 26). Hence, we expressed all *A. coerulea CYP716*s in yeast together with *GgBAS*. GC–MS analysis of spent medium showed new peaks for yeasts expressing *CYP716A110* and *CYP716A111*, as well as *CYP716A113v1* (Fig. 3c). For the latter, these peaks matched those of *CYP716A113v1* expressed in yeast without any heterologous OSC, thus representing the same nonspecific reaction products. The new peaks from the yeast expressing *CYP716A111* matched the NMR-verified 16β-hydroxy β-amyrin standard (Supplementary Fig. 1d), suggesting that CYP716A111 can catalyse 16β-hydroxylation of β-amyrin like *P. grandiflorus* CYP716A141. GC–MS analysis of the medium from the yeast expressing *CYP716A110* revealed new peaks corresponding to the erythrodiol and oleanolic acid standards (Supplementary Fig. 1d), hence uncovering another three-step C28 oxidase CYP716 representative. Semi-quantitative analysis of the accumulated oleanane-type triterpenoids in the latter strains showed that CYP716A110 and CYP716A111 display less activity in terms of relative substrate consumption and product concentrations as compared with the CYP716s of *C. asiatica* and *P. grandiflorus* catalysing the same reactions (Table 1 and Supplementary Table 3).

In summary, we have shown that *A. coerulea* possesses CYP716 enzymes that can oxidize triterpenoids of the cycloartane and oleanane types.

**Kingdom-wide phylogenetic analysis of the CYP716 family**. To shed light on the evolution of the CYP716 family, we conducted an extensive phylogenetic analysis, with the set of characterized CYP716 sequences complemented with a comprehensive plant kingdom spanning set of publicly available full-length CYP716 sequences retrieved from the Cytochrome P450 Homepage (http://drnelson.uthsc.edu/cytochromeP450.html), the NCBI sequence database (www.ncbi.nlm.nih.gov/), the 1KP initiative[36] (https://sites.google.com/a/ualberta.ca/onekp/), the PhytoMetaSyn

medicinal plant transcriptomics resource[24] and the PLAZA comparative genomics resource[37]. The generated data set included 409 sequences from 205 plant species, which were all systematically named (Supplementary Data set 1). The oleanane-metabolizing CYP88D6 (ref. 38), which also belongs to clan 85 of plant P450s and represents the functionally characterized triterpenoid-metabolizing P450 that is most closely related to the CYP716 family, was chosen as outgroup.

The resulting tree clearly depicts the CYP716 family as a congruent entity that distinguishes itself from its closest relatives, the CYP718 and CYP728 families, which were also included in the phylogenetic analysis (Fig. 4a and Supplementary Fig. 4). Traditionally, P450s have been divided into families and subfamilies based on >40 and >55% sequence homology thresholds, respectively. These rules were also used to define the borders of the family, although some members that failed to meet the 40% requirement were still included as CYP716 members because of a clear phylogenetic relation and were therefore also assigned a CYP716 nomenclature. The phylogenetic tree displays several clusters of related sequences that often, but not strictly, consist of members of a single subfamily. These are from now on referred to as 'subgroups' and have probably emerged at different points during evolution (Fig. 4b). Based on the tree topology and the taxonomic occurrence, the subgroups could be divided over three evolutionary categories that we named the 'Eudicot', 'Angiosperm' and 'Ancient' CYP716s (Fig. 4a and Supplementary Fig. 4).

Members of the 'Eudicot' subgroups (or subfamilies) are found exclusively in eudicots and many subgroups are found already in the earliest eudicot orders. Notably, this category includes all the biochemically characterized P450s. Members of the 'Angiosperm' subgroups contain CYP716s from several clades such as Magnoliids, Chloranthales and Amborellales, but also CYP716s from eudicot species. Finally, the 'Ancient' subgroups exclusively contain CYP716s from pre-angiosperm plants, namely gymnosperms, ferns, lycopods, mosses and liverworts.

The characterized members of the 'Eudicot' category are divided over all five subgroups (A, E, Y, S and C). The by far biggest and most common CYP716 subgroup is the very well-conserved group A. The vast majority of the characterized enzymes from this subgroup are C-28 oxidases and, conversely, this subgroup comprises all C-28 oxidases hitherto reported[9,10,16,20,26,39] and identified in this study, except for the multifunctional *P. grandiflorus* CYP716A141 (Supplementary Fig. 4). Other known activities in this subgroup include the C-3 oxidation by *Artemisia annua* CYP716A14v2 (ref. 11), the 22α-hydroxylation of α-amyrin by *Arabidopsis thaliana* CYP716A2 (ref. 12), the unknown hydroxylation of tirucalla-7,24-dien-3β-ol by *A. thaliana* CYP716A1 (refs 8,12) and the 16β hydroxylation of β-amyrin and putative hydroxylation of cycloartane by *A. coerulea* CYP716A111 and CYP716A113v1, respectively. Except for CYP716A14v2, enzymes from this subgroup seem to target the same region on the triterpenoid backbone around the C-28 β-methyl group on a (pentacyclic) triterpenoid backbone. Strikingly, subgroup A members were found in every eudicot with a complete genome available and, in general, seemed to be present from basal eudicot orders, such as the Ranunculales, to the furthest diverged orders, such as the Brassicales.

Although not as ubiquitous as the subgroup A, also the other 'Eudicot' CYP716 subgroups, that is, E, Y, S and C, are widely distributed over eudicots. Subgroup E is a rather large group and includes CYP716E41, the oleanane C-6β hydroxylase discovered here (Supplementary Fig. 4). Subgroup Y includes the previously reported C-16α hydroxylase CYP716Y1 from *Bupleurum falcatum*[19] and the dammarane C-12β hydroxylase CYP716U1

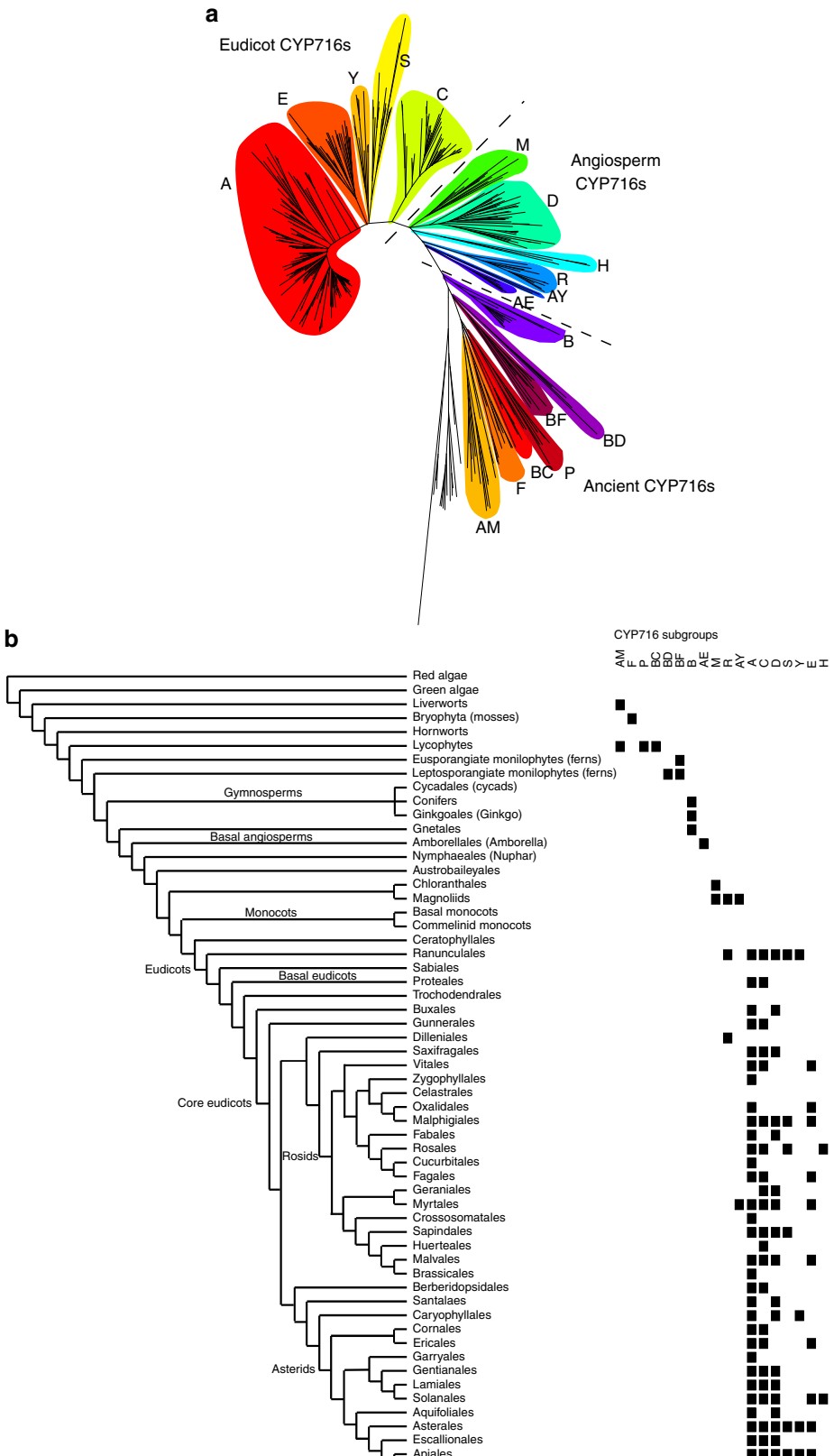

**Figure 4 | Maximum likelihood phylogenetic analysis of the CYP716 family.** (**a**) CYP716 members form distinct subgroups and can be divided in eudicot, angiosperm and ancient CYP716s. (**b**) CYP716 subgroups distributed per plant order, showing the point of emergence of different subgroups and suggesting diversification of the CYP716 family in early eudicots. Plant orders are depicted according to the APG III system (angiosperms)[21] and the tree of life project (the rest, https://tree.opentreeoflife.org/) taxonomy.

(renamed from CYP716A47) from *Panax ginseng*[18], as well as the multifunctional *C. asiatica* CYP716A141 enzyme with C-16β hydroxylase and C-28 oxidase activity found in this study. Subgroup S contains CYP716S1 (ref. 17) (renamed from CYP716A53v2), a dammarane C-6α hydroxylase from *P. ginseng* and the oleanane 12,13α-epoxidase CYP716S5 characterized here. Finally, subgroup C is again a larger group found already in basal eudicots that includes the *C. asiatica* C-2α hydroxylase CYP716C11 from this study. Members of all eudicot CYP716 subgroups, except E, can be found already in the earliest diverged basal eudicots, that is, the Ranunculales, suggesting that the diversification of the family has occurred early in the eudicot lineage. The emergence of group E can be placed in the core eudicots or earlier (Fig. 4b). All five eudicot subgroups are present in multiple orders spanning most eudicots, but nonetheless analysis of sequenced plants indicates that members of the E, Y, S and C subgroups have a non-universal distribution in extant species, contrary to the CYP716A members, and thus most probably reflect plant clade-specific evolution rather than conservation.

The 'Angiosperm' category does not contain any characterized CYP716 enzymes. The mutually very closely related subgroups M and D contain CYP716s from Magnoliids (and Chloranthales) and eudicots, respectively. Although they both emerged earlier than the 'Eudicot' CYP716s, as judged from the high sequence similarity (43–59% and 45–55% similarity to CYP716A12), they are most probably the closest relatives to the 'Eudicot' CYP716s. Other 'Angiosperm' CYP716s comprise the H (found in Solanales and Fabales), R (found in mixed orders), AY (mixed orders) and AE (found in Amborellales) subgroups. Subgroup R spans the evolutionary distance from the Magnoliids to the Dilleniales (eudicots) and includes the still uncharacterized *A. coerulea* members CYP716R5v1 and CYP716R5v2.

The 'Ancient' CYP716s comprise the subgroups B (found in gymnosperms), BD (found in leptosporangiante monilophytes; ferns), BF (found in eusporangiante monilophytes and leptosporangiante monilophytes; ferns), P (found in lycopods), BC (found in lycopods), F (found in mosses) and AM (found in liverworts and lycopods). Some of these sequences fail to meet the >40% amino acid sequence identity criterium but are clearly separated from the closest non-CYP716 relatives, that is, the CYP728 and CYP718 P450s, and may thus be regarded CYP716s.

Although CYP716s could be found in nearly all land plant orders, curiously only three *CYP716* sequences from two monocot plants, *Phormium tenax* (Asparagales) and *Xerophyllum asphodeloides* (Liliales) (PteCYP716AX1, PteCYP716A192 and XaCYP716A193; Supplementary Data set 1) could be detected across all the mined databases. Considering that these databases together cover over 100 monocot plants that represent most of the monocot orders, many of which have their full genome sequenced, we regarded these three putative monocot CYP716s as anomalies. We postulate that the CYP716s have evolved early during evolution, because they were present in the very first land plants, mosses and liverworts, but have been lost during evolution in monocots, because they were absent in all (but two) species of this branch of the plant kingdom.

## Discussion

P450 enzymes are known for the diverse reaction mechanisms they employ in producing an enormous variety of plant metabolites. Here we focused on the CYP716 family, of which we cloned 22 members from the medicinal plant species *C. asiatica*, *P. grandiflorus* and *A. coerulea*, and expressed them in a previously described triterpenoid-producing yeast system[19]. This analysis resulted in the discovery of as many as ten hitherto unreported triterpenoid oxidation activities with different backbones as substrates and at different positions of the backbones, which showcased the functional versatility of the CYP716 enzymes and their important role in and specific evolvement towards plant triterpenoid biosynthesis.

This is particularly well exemplified by the fact that the *C. asiatica* CYP716 enzymes characterized in this study cover three of the four oxidative decorations of pentacyclic triterpenoid backbones known to exist in *C. asiatica* oleanane and ursane saponins[23], that is, C-2α, C-6β and C-28 oxidation, with only the enzyme catalysing C-23 hydroxylation yet to be discovered.

Likewise, in addition to the more common β-amyrin oxidation reactions catalysed by the *P. grandiflorus* CYP716A140 (three-step C-28 oxidation) and CYP716A141 (C-16β-hydroxylation and three-step C-28 oxidation), CYP716S5 was found to perform 12,13α epoxidation, thereby expanding the repertoire of CYP716-catalysed reactions with an interesting function. Indeed, combining this activity with the three-step C-28 oxidation activity of CYP716A140 allowed generating the heterocyclic triterpenoid 12α-hydroxy β-amyrin-13,28β-lactone, highlighting a function that was hitherto not reported for triterpenoid-metabolizing enzymes.

Finally, the CYP716 enzymes are known to be involved in the biosynthesis of triterpenoids with oleanane, ursane, lupane and dammarane backbones. The *A. coerulea* CYP716 enzymes illustrate the range of different triterpenoid backbones that can be accepted, with CYP716A113v1 able to oxidize the tetracyclic triterpenoid cycloartenol and possibly other tetracyclic triterpenoids such as yeast sterol precursors, and CYP716A110 and CYP716A111 able to oxidize the pentacyclic β-amyrin. This further underscores that CYP716s are capable of (specifically) using different OSC products, a capacity that seems to have emerged early during evolution.

Phylogenetic analysis suggests that CYP716 enzymes have emerged in the first land plants, in which they may have a yet unknown role in triterpenoid metabolism. In eudicots, neofunctionalization of the CYP716 family appears to have had a big role in the evolution of the triterpenoid and triterpene saponin diversity. In monocots, the CYP716 family has been lost and may perhaps be replaced by another family (or families), such as the atypical CYP51H[40,41], to account for the existing diversity in monocot triterpenoid structures.

Accordingly, compared with other P450 families that are known to have members involved in eudicot triterpenoid biosynthesis, such as CYP72A[7,42] and CYP88D[14,38], the CYP716 family seems to have more of a universal than a plant species-, genus- or family-specific role in triterpenoid biosynthesis[14]. For instance, CYP716 subgroup A members are found in virtually all eudicots and are to date the only P450s known to perform C-28 oxidation of pentacyclic triterpenoids (except for the multifunctional subgroup Y member CYP716A141 from *P. grandiflorus*), the by far most common decoration found in saponins. Furthermore, a C-28 carboxy group was found to be a prerequisite for further decorations of the triterpenoid backbone, for instance by the *C. asiatica* CYP716E41 and CYP716C11 enzymes in this study and the *M. truncatula* non-CYP716 enzymes CYP72A67 and CYP72A68v2 (ref. 7), suggesting a seminal role for this activity and an earlier employment (from an evolutionary point of view) of a CYP716A in the biosynthetic pathways in which the latter enzymes are involved.

Evolutionarily, neofunctionalization of the CYP716s may have coincided with the emergence of eudicot OSCs with specific roles in specialized metabolism, thus diverging from the OSCs delivering the precursors for primary sterol metabolism. Triterpenoid backbone types can be divided into two classes according to the mechanism of their cyclization, that is, derived

from the OSC reaction intermediate protosteryl cation or its stereoisomer dammarenyl cation. Primary metabolism-related triterpenoids, such as the sterols, arise solely from the protosteryl cation but the most common, specialized metabolism-related triterpenoids, such as the oleananes, ursanes, lupanes and dammaranes, arise from the dammarenyl cation[4,43]. The OSCs employing the latter mechanism have been proposed to have followed at least two distinct evolutionary paths, either starting from an ancient lanosterol synthase-like OSC in the case of eudicots or an ancient cycloartenol synthase in the case of monocots[44]. Hence, the difference in CYP716 occurrence between monocots (no CYP716s) and eudicots (diverse CYP716 subfamilies and functions) probably reflects the divergent evolution of the triterpenoid biosynthesis machinery in these two clades. The emergence of ancient lanosterol synthase-like OSCs may have been a driver in the evolution and diversification of CYP716s in eudicots. Dammarenyl cation class OSCs have not yet been reported in plants of more ancient descent than the monocots and, apart from a few sporadic examples[45,46], the typical products of such OSCs are generally absent in pre-monocot species. This is exemplified by conifers, which produce only cycloartane and/or lanostane-type triterpenoids[47].

With regard to the CYP716s that are not active in our yeast system, it remains to be determined what contribution they may make to triterpenoid biosynthesis. Lack of detected activity for a particular CYP716 family member may have different causes. It is possible that it metabolizes different terpenoid backbones, from sterols over other (tetracyclic) triterpenoids to any other class of terpenoid. For instance, as the CYP716s may only work on a downstream triterpenoid pathway intermediate, thus requiring the prior activity of other P450 enzymes, we may miss detection of triterpenoid metabolizing activity in our current setup. It is noteworthy though that most of the CYP716s for which we could not find a function have been cloned from A. coerulea gDNA. Hence, it cannot be excluded that some of these genes may not be expressed and thus represent pseudogenes. Furthermore, as we are expressing the plant CYP716s in a yeast system, it cannot be excluded that the enzymes are not expressed and/or folded in a correct way, thus being inactive in this heterologous host.

In conclusion, the CYP716 enzymes discovered and characterized here contribute to our knowledge about triterpenoid biosynthesis in different plants and show the potential of the CYP716 family enzymes as a source of new functionalities. They are also an addition to the toolbox of available enzymes for metabolic engineering and heterologous production of valuable triterpenoids. The modular biosynthesis of triterpenoids makes P450s ideally suited for mix-and-match combinatorial biochemistry and other synthetic biology programmes in heterologous organisms such as S. cerevisiae[19,48,49]. Many recent promising studies have shown saponins and triterpenoids to have commercial potential[2]. Discovery of new enzymes, in particular those activating new positions on the carbon backbones and enabling further biological and synthetic manipulation of triterpenoids, paves the way to the expansion of the repertoire of triterpenoid compounds available for bioactivity assays and, ultimately, to the development of new drugs or other high-value chemicals.

## Methods

**Gene discovery.** Publicly available transcriptome resources were screened for candidate CYP716 sequences by TBLASTX searches using the nucleotide sequence of M. truncatula CYP716A12 (ref. 26). Sequences with >40% amino acid identity were retained as candidates (Supplementary Table 2). For C. asiatica, four sequence assemblies were obtained from the PhytoMetaSyn project[24] and screening for CYP716 sequences yielded five full-length candidates and one incomplete sequence. For P. grandiflorus, raw 454 pyrosequencing data were obtained from 'The Compositae Genome Project' (http://compgenomics.ucdavis.

edu/). TBLASTX searches in the raw sequencing data yielded six incomplete CYP716 candidates. Full-length coding sequences of five of the six P. grandiflorus candidates were obtained via RACE PCR. The A. coerulea genome data were obtained from the Phytozome portal (http://phytozome.jgi.doe.gov/) and screening for CYP716 sequences yielded 18 full-length candidates. Because of high sequence similarity between some of the CYP716 sequences, only 12 of them were cloned for further analysis.

**Plant material and cloning template preparation.** P. grandiflorus (purchased from www.johnsons-seeds.com) and A. coerulea 'Origami Red' (purchased from www.mr-fothergills.co.uk) seeds were germinated in soil and plants were grown in a 16 h/8 h day/night regime at 20 °C. Before sampling, 1-month-old plants were sprayed with 1 mM of methyl jasmonate (dissolved in deionized water with 0.1% (v/v) Tween-20) and kept in a plastic bag for 24 h. For both species, RNA was prepared from roots and leaves with the RNeasy kit (Qiagen) and used for cDNA preparation with the iScript kit (BIO-RAD). The RACE template was prepared from RNA using the SuperScript II 5X first-strand buffer (Invitrogen), 0.025 M dithiothreitol, 2.5 mM dNTPs, iScript reverse transcriptase (BIO-RAD) and 0.25 mM primer RACE3 (Supplementary Table 4). A. coerulea gDNA was extracted as reported[50]. For C. asiatica, the cloning template consisted of an Uncut Nanoquantity cDNA library (custom-made by Invitrogen) generated from RNA derived from greenhouse-grown methyl jasmonate-treated C. asiatica plants.

**Cloning of CYP716 candidates and yeast expression vectors.** The full-length CYP716-coding sequences, the M. truncatula cytochrome P450 reductase MTR1 (Medtr3g100160) and the S. lycopersicum cycloartenol synthase (SlCAS) were PCR amplified (for primers, see Supplementary Table 4) and Gateway recombined into the donor vector pDONR207 or pDONR221. A. coerulea CYP716 FL-ORFs were amplified from either cDNA (CYP716A105, CYP716A107, CYP716A109, CYP716A110, CYP716A113v1, CYP716R5v2 and CYP716A114) or gDNA (CYP716A100, CYP716A102, CYP716A103, CYP716A111 and CYP716A112) (Supplementary Table 5). In the latter case, the candidate genomic sequences were amplified and ligated into the pJET1.2 blunt cloning vector (Thermo Scientific). The three exons of each candidate were amplified separately from these plasmids before the FL-ORFs were finally amplified in a PCR reaction containing the three exon fragments and the uttermost forward and reverse primers. Constructs encoding the self-processing polyproteins CYP716A83 with CYP716C11, GgBAS[19] with MTR1 and CaDDS[19] with MTR1 were designed using the T2A viral peptide. The coding sequences to be joined were separately amplified (for primers, see Supplementary Table 4) with a partial T2A overhang and a Type IIS restriction site. The resulting PCR fragments were digested and ligated together using T4 DNA ligase (Thermo Scientific) and Gateway recombined into the donor vector pDONR221.

For cloning convenience, the pESC-URA-tHMG1-DEST plasmid was generated by inserting a Gateway cassette into the pESC-URA[GAL10/tHMG1][19] plasmid. The Gateway cassette was amplified from pDEST14 (Invitrogen) using primers pESC-DEST1 and pESC-DEST2 (Supplementary Table 4) and cloned into pJET1.2 (Thermo Scientific) for sequence verification. The Gateway cassette was subsequently excised from pJET1.2 by digestion with BclI and NheI. Simultaneously, the plasmid pESC-URA[GAL10/tHMG1] was linearized using BamHI and NheI. Both the insert and the linearized vector were gel purified and ligated using T4 DNA ligase (Invitrogen) according to the manufacturer's instructions. The resulting destination vector pESC-URA-tHMG1-DEST was sequence verified before use.

To generate yeast expression vectors, sequence-verified entry clones were Gateway recombined into the high-copy number yeast destination vectors pESC-URA-tHMG1-DEST, pAG423GAL-ccdB (Addgene plasmid 14149 (ref. 51)), pAG424GAL-ccdB (Addgene plasmid 14151 (ref. 51)) or pAG425GAL-ccdB (Addgene plasmid 14153 (ref. 51)).

**Generation of plasmids for CRISPR/Cas9 in yeast.** A plasmid for CRISPR/Cas9 in yeast that contains both Cas9 and a gRNA cassette according to the system reported by DiCarlo et al.[52] was generated. The Cas9 expression cassette was PCR amplified from p414-TEF1p-Cas9-CYC1t (Addgene plasmid 43802) using primers combi3244 and combi3247 (Supplementary Table 4), each containing a HpaI restriction site at the 5′-terminus. The resulting fragment was cloned into pJET1.2 and sequence verified. Subsequently, the Cas9 cassette was cut out using HpaI, gel purified and cloned into the vector backbone of a PvuII-treated, dephosphorylated and gel-purified pESC-URA plasmid (Agilent). The resulting plasmid was named pCAS1. Next, a Gateway cassette was PCR amplified from pDEST14 (Invitrogen) using primers combi1715 and combi2287 (Supplementary Table 4). The PCR fragment was treated with XbaI and NheI, gel purified and cloned into the NheI-linearized and dephosphorylated pCAS1, yielding pCAS-ccdB.

**Generation and cultivation of yeast strains.** Yeast strain PA14 was derived from strain TM1 (ref. 19) by knocking out the TRP1 gene using CRISPR/Cas9. The TRP1 knockout construct was generated by PCR amplification of SNR52p and sgRNA-CYC1t from p426-SNR52p-gRNA.CAN1.Y-SUP4t (Addgene plasmid

43803) using primers combi3245 and crispr014, and combi3246 and crispr031, respectively (Supplementary Table 4). The individual fragments were fused by overlap extension PCR, subcloned into pDONR221, sequence verified and finally Gateway recombined into pCAS-ccdB, yielding pCAS-TRP1. Subsequently, 200 ng of pCAS-TRP1 and 10 μmol of double-stranded DNA (prepared by annealing the single-stranded oligonucleotides crispr059 and crispr060; Supplementary Table 4) as homologous recombination donor were co-transformed in yeast strain TM1. The resulting colonies were analysed for positive CRISPR events by replica plating on SD medium (Clontech) with or without tryptophan. Tryptophan auxotrophs were further confirmed by Sanger sequencing. Auxotrophic strains were cured of pCAS-TRP1 by counter-selection on plates containing 1 mg ml$^{-1}$ 5-fluoroorotic acid (Zymo Research) and the resulting TM1-derived trp1 strain was named PA14 (Supplementary Table 6).

All yeast strains used in this study (Supplementary Table 6) were generated from strain TM1 or PA14 and were cultivated in the presence of MβCD, to enhance triterpenoid production and enable sequestering triterpenoids from the spent medium, as previously described[19].

**Metabolite extractions and profiling.** The spent yeast culture medium was extracted twice with 0.5 volumes of hexane and once with 0.5 volumes of ethyl acetate, the resulting organic extracts were pooled, evaporated to dryness and trimethylsilylated for GC–MS analysis. Trimethylsilylation was carried out with 50 μl of N-methyl-N-(trimethylsilyl)trifluoroacetamide (Sigma-Aldrich) and 10 μl of pyridine. GC–MS analysis was carried out as previously described[19]. For yeast strain 37, metabolites were extracted thrice with 0.5 volumes of ethyl acetate and further processed and analysed by LC-APCI-FT-ICR-MS as previously described[53]. For semi-quantitative GC–MS analysis of enzyme products, four biological replicates of each yeast strain were cultivated and extracted as indicated above. For this analysis however, cholesterol was spiked as an internal standard to each sample before extraction. Integration of peak area values from extracted ion chromatograms of representative ions for known triterpenoids was conducted with the ChemStation Software (Agilent). For the calculations, the peak area values for each compound were normalized using the internal standard.

**Preparation of 6β-hydroxy maslinic acid and 16β-hydroxy β-amyrin.** For preparation of 6β-hydroxy maslinic acid, a 3-l culture of yeast strain 14 (Supplementary Table 6) was cultivated in the presence of MβCD as previously described[19] and extracted thrice with 200 ml of ethyl acetate. The organic phases were pooled and evaporated to dryness. 6β-Hydroxy maslinic acid was washed thrice by resuspension of the residue into 1 ml of ethyl acetate and subsequent precipitation by adding 4 ml of hexane and removal of the solvent. Next, 6β-hydroxy maslinic acid was purified by column chromatography using 20 ml of silica gel as stationary phase and increasing concentrations of diethyl ether in hexane as mobile phase. After loading the extract, the column was washed with 50 ml of 10% diethyl ether in hexane and elution of 6β-hydroxy maslinic acid was achieved using 20% diethyl ether in hexane. Using thin layer chromatography, fractions containing 6β-hydroxy maslinic acid were identified, after which they were pooled, evaporated and subjected to NMR analysis.

16β-Hydroxy β-amyrin was purified from a 1-l culture of yeast strain 32 (Supplementary Table 6) grown and induced as previously described[19]. The spent medium was extracted thrice with 200 ml of hexane that was pooled and evaporated to dryness. Subsequently, the compound was loaded onto a 10-ml silica column for purification. The column was eluted with 10% ethyl acetate in hexane and fractions were collected and screened for the presence of 16β-hydroxy β-amyrin with thin layer chromatography. Fractions containing 16β-hydroxy β-amyrin were pooled, evaporated to dryness and subjected to NMR analysis.

**NMR analysis.** The NMR spectra of purified 16β-hydroxy β-amyrin, 6β-hydroxy maslinic acid and maslinic acid standard were measured on an Avance II Bruker spectrometer operating at a $^1$H frequency of 700 MHz and equipped with a 5-mm $^1$H/$^{13}$C/$^{15}$N TXI-z probe. The dry samples were dissolved in MeOD-d4 (99.96% D), to have an interference of the solvent signal as low as possible. In addition to boost the sample concentration, a MeOD-d4 shigemi tube was used for the 6β-hydroxy maslinic acid and corresponding standard samples. All spectra were referenced to the residual solvent-signal at 3.31(5) p.p.m. for the $^1$H frequency and 49.15(7) p.p.m. for the $^{13}$C frequency. The spectra recorded on the samples included 1D $^1$H, 2D $^1$H-{$^1$H} correlation spectroscopy (COSY), $^1$H-{$^1$H} total correlation spectroscopy (TOCSY) (100 ms spinlock), $^1$H-{$^{13}$C} heteronuclear single quantum coherence (HSQC), two $^1$H-{$^{13}$C} heteronuclear multiple bond correlation (HMBC) with 8 or 4 Hz long-range coupling constants and $^1$H-{$^1$H} off-resonance rotating frame nuclear Overhauser effect spectroscopy (ROESY) (300 ms spinlock). No $^{13}$C spectra were recorded due to the low available sample amount or an assignment of all carbon atoms using the heteronuclear single quantum coherence (HSQC) and heteronuclear multiple bond correlation (HMBC) spectra. All spectra were processed using TopSpin 3.5 pl1 or TopSpin 3.2 pl6 software. More details are provided in Supplementary Methods.

**Phylogenetic analysis.** Over 400 putative CYP716 amino acid sequences from over 200 plant species spanning the whole plant kingdom (Supplementary Data set 1) were collected from public databases, namely the Cytochrome P450 Homepage (http://drnelson.uthsc.edu/cytochromeP450.html), the PhytoMetaSyn medicinal plant transcriptomics resources[24], Phytozome (http://phytozome.jgi.doe.gov) and 1KP transcriptome resources[36] with at least 35% amino acid sequence identity to CYP716A12 (ref. 26) and with a complete sequence of at least 98% as criteria. Selected non-CYP716 triterpene-metabolizing P450s were picked from NCBI (http://www.ncbi.nlm.nih.gov). The sequences were aligned using the MUSCLE algorithm[54]. ProtTest[55] was run to find the optimal amino acid substitution model. The phylogenetic tree was built using the MEGA5 software[56] with the JTT + GAMMA + I + F model.

**Data availability.** All relevant data are available from the authors. Sequence data from this study can be found in the GenBank/EMBL databases under the following accession numbers: CYP716A86, KU878848; CYP716A83, KU878849; CYP716D36, KU878850; CYP716E41, KU878851; CYP716C11, KU878852; CYP716A140, KU878853; CYP716S4, KU878854; CYP716A141, KU878855; CYP716S5, KU878856; CYP716S6, KU878857; CYP716A100, KU878858; CYP716A102, KU878859; CYP716A103, KU878860; CYP716A105, KU878861; CYP716A109, KU878862; CYP716A107, KU878863; CYP716A110, KU878864; CYP716A112, KU878865; CYP716A113v1, KU878866; CYP716R5v2, KU878867; CYP716A114, KU878868; MTR1, KU878869; and CYP716A111, KY047600.

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

## Acknowledgements

We thank Annick Bleys for help with preparing the article. This work was supported by the European Union Seventh Framework Program FP7/2007-2013 under grant agreement number 613692-TriForC. J.P. and J.M. are post- and predoctoral fellows of the Research Foundation Flanders, respectively. P.A. and T.M. are indebted to the VIB International Fellowship Program and the Special Research Funds from Ghent University for a pre- and postdoctoral fellowship, respectively. R.C. and S.S. thank Science Campus Halle for support.

## Author contributions

K.M., J.P. and A.G. conceived the study and designed experiments. K.M., J.P., D.B., P.A., R.C., S.S., T.M. and J.M. performed experiments. K.M., J.P., D.B., J.M., A.A., D.R.N. and A.G. analysed experiments. K.M., J.P. and A.G. wrote the manuscript with support from all authors.

## Additional information

**Competing financial interests:** The authors declare no competing financial interests.

