## [Peer Review File · Nature Communications]

Reviewers' comments:

Reviewer #1 (Remarks to the Author):

This manuscript provides a broad view of the role of the CYP716 family of cytochrome P450 mono-oxygenases in plant triterpenoid biosynthesis. The authors have taken an interesting alternative approach to the role of CYPs in natural products biosynthesis. Instead of focusing on a particular pathway, here they focus on this CYP716 family, which seems to be exclusively involved in plant triterpenoid biosynthesis, an area of particular interest to this group, and simply asked how far this type of role can be extended. Through this novel approach, they uncover a variety of novel activities beyond the oxygenation/oxidation of C28 of the pentacyclic amyrin/lupeol from a methyl to carboxylic acid that typifies the large CYP716A sub-family, with such alternative activity perhaps highlighted by production of a lactone ring. Not accidentally, these results further provide some insight into the biosynthesis of various triterpenoids of medicinal interest, although the authors do characterize the family across a wider phylogenetic range of plants. Indeed, they find that an early diverging eudicot *Aquilegia coerulea* has several-fold more CYP716 family members than the investigated medicinal herbs from the Asterids. The biochemical analyses reported here are by no means comprehensive, with only a few triterpene backbones examined for the CYP716 family members from each plant, biased by the triterpenoids of interest from that particular plant. It is thus perhaps not surprising that activity was found for only approximately a third of the investigated CYP716 family members. However, this does leave in question what contribution the remaining two-thirds (majority) of these CYPs make to triterpenoid biosynthesis. For example, it might then be possible that some of these instead act in other types of natural products biosynthesis (e.g., of sester- or di-terpenoids, much as CYP88 family members have been diverted from gibberellin diterpenoid metabolism to triterpenoid biosynthesis). Accordingly, it seems a bit presumptuous to call the CYP716 family the "cradle of triterpenoid diversity", while it certainly seems to have contributed, the title perhaps overstates the case by some measure. Nevertheless, this manuscript does present an interesting and novel approach to examining the role of CYPs in natural products biosynthesis, and the authors should be applauded for their bold study.

Reviewer #2 (Remarks to the Author):

Conceptually, the manuscript by Miettinen et al. addresses the very interesting prospect that cytochrome P450 enzymes mediate the secondary diversification of the amyrin, oleanane, ursane and cycloartane triterpene scaffolds in plants. Although not explicitly stated in the manuscript, the notion seems to be that progenitor P450s with specificity for tetracyclic triterpenes could have given rise to those with pentacyclic specificity, then new alleles neofunctionalized with different regio- and successive oxidation reactions. The authors approach has been to screen 3 plants known for their diversity of oxidized triterpenes for their CYP716 family of CYPs, to functionally characterize these by expression in yeast engineered for amyrin, oleanane, ursane or cycloartenol production, then to chemically profile the yeast for new triterpene products. Finally, the authors appear to use a very common clustering algorithm to suggest some affiliation of CYP716s to particular plant families and lineages.

But does this analysis really establish that the CYP716s are the cradle of chemical diversity of triterpenes?

There are many experimental components to this work and all are expertly performed. First, they have used a variety of transcriptomic resources to develop databases to screen for the respective CYP716 transcripts. In some cases, they had to generate sequence information themselves and in

other cases they needed to generate full-length clones for the desired transcripts identified in the RNA sequence information. They then had to introduce the various CYP genes into yeast producing the various triterpene scaffolds, then profile the yeast extracts by GC-MS, and in at least one case, use LC-MS to identify a temperature labile lactone structure.

There are a few issues the authors may want to consider.

First, the chemical profiles as presented are very aesthetically pleasing, but almost impossible to read. As much as I like these plots, I really want to see more quantitative information and suppose a carefully crafted table could do that, with the chromatograms provided in the supplementary data. I do mean quantitative information because I would like to sum up the total of all the intermediates to gain a pseudo substrate to product ratio. For example, in the control BAS expressing line, 100 units of beta-amryin. Then in the line expressing BAS plus CYP716A86 or A83, I would expect the total triterpene content to be about the same, 100 units, A83 having much less beta-amryin relative to A86. If this were not the case, then the authors might have some explaining to do. I'd also like to know that these accumulation patterns are reproducible - 3 replicates within an experiment and certainly with the same results observed in subsequent experiments.

The quantitative analysis is also important when it comes to offering the maps for catalytic steps, as in Figure 2. When the authors observe that CYP enzymes can work in either order of catalysis, it might be helpful to know if the quantitative yields varied for the initial step. That could be indicative of a preferred order in the cascade.

My second concern pertains to the phylogenetic inferences. I'm first assuming the trees are neighbor joining and derived from compiled sequence alignments. Given that the CYP716 family is defined by be greater than 40% identical, the clustering observed could be related to function, but it would be nice if the authors could elaborate on this point. For example, commenting about which members within the trees exhibit different or promiscuous triterpene scaffold preference, followed by regio- and successive oxidation specificities. The cladograms are probable sufficient for inferring general origins, but even that analysis isn't very well described. Given that the authors want the general reader to take away that these CYPs are the cradle of chemical diversity for triterpenes, improving the sophistication and in depth considerations of this part of the work would certainly be welcomed.

Overall this is a very fine manuscript. What I'm suggesting is some more critical consideration of the data and its presentation in order to really drive home the points the authors wish to make.

Reviewer #3 (Remarks to the Author):

This paper demonstrates how powerful can be the mining of the genomic information that recently became available for discovering new cytochrome P450 catalytic functions. Targeted mining and functional analysis of the CYP716 family members in just three medicinal plant species allows the authors to describe the enzymes catalyzing ten unreported terpenoid oxidation activities. Most reaction products are identified, demonstrating, among others, the formation of a triterpenoid lactone from an epoxide intermediate. One of the targeted plants, *Aquilegia coerulea*, is representative of the basal eudicots. Information on its CYP716s catalytic activity, together with in depth phylogenetic analysis and database mining provides interesting insight into the CYP716 family functional diversification along land plant evolution. Work is well done and text clearly written. Comments of P450 phylogeny essentially appropriate.

Minor comments:

- page 9 line 2: I assume that the authors mean an m/z of 686 and not 586, since it would not make sense.
- yeast-dependent cycloartenol metabolism has been documented on several occasion and this needs to be mentioned (see Gas-Pascual et al PLOS one, 9, (2014)).
- it is not so clear what the authors mean by "earlier diverged CYP716s", after referring to Amborellales. Do they mean non-existent genes/enzymes?
- pages 16 and beyond: the two monocot plants showing the presence of CYP716 gene(s) are both Liliales. This could be worth mentioning, if possible in a broader functional or evolutionary context.
- it would be worth mentioning, at least in the method section, why MC β D is added to the medium.
- figures: all of them are extremely small and not quite readable. Especially carbon numbering of the formula is not readable at all. Also subfamily names on the right part of figure 4b.

REVIEWERS' COMMENTS:

Reviewer #1 (Remarks to the Author):

This authors have adequately addresses my previously expressed concerns, particularly with the new title and discussion of potential rationales for the many CYP716 family members that do not exhibit activity in the somewhat limited assays that are currently reported.

Reviewer #2 (Remarks to the Author):

The authors have fully and adequately responded to my previous concerns. I'm especially appreciative of the extra effort given to providing the quantitative information requested in new Table 1. Only one minor suggestion - it would be informative for the general reader if the method used for the phylogenetic tree (maximum likelihood) was stated in the figure legend. The title of the supplementary materials should also be changed to the new title of the manuscript. The new title is actually much better and informative in this reviewer's opinion.

Reviewer #3 (Remarks to the Author):

I found proper answers to all my minor comments.

I must however express my skepticism concerning the tentative quantification required by reviewer 2. Cytochromes P450, even when closely related, show very different levels of expression in yeast. This is especially true for fast evolving families to which the CYP716 family belongs. In addition, their respective levels of expression in vivo is impossible to quantify. Authors state themselves to their answer to reviewer 1 that they cannot evaluate if a P450 is expressed or not in their whole yeast assay.

Consequently, it is extremely risky to draw quantitative and comparative activity conclusions from experiments carried out using whole yeast assays, as done in this updated work. Personally, I would have refused to compell to Reviewer 2 request for quantitative evaluation, since to my eyes it has very limited chances to be be reliable.

Response to the Reviewers

Reviewer #1:

*1/ This manuscript provides a broad view of the role of the CYP716 family of cytochrome P450 monooxygenases in plant triterpenoid biosynthesis. The authors have taken an interesting alternative approach to the role of CYPs in natural products biosynthesis. Instead of focusing on a particular pathway, here they focus on this CYP716 family, which seems to be exclusively involved in plant triterpenoid biosynthesis, an area of particular interest to this group, and simply asked how far this type of role can be extended. Through this novel approach, they uncover a variety of novel activities beyond the oxygenation/oxidation of C28 of the pentacyclic amyirin/lupeol from a methyl to carboxylic acid that typifies the large CYP716A sub-family, with such alternative activity perhaps highlighted by production of a lactone ring. Not accidentally, these results further provide some insight into the biosynthesis of various triterpenoids of medicinal interest, although the authors do characterize the family across a wider phylogenetic range of plants. Indeed, they find that an early diverging eudicot *Aquilegia coerulea* has several-fold more CYP716 family members than the investigated medicinal herbs from the Asterids. The biochemical analyses reported here are by no means comprehensive, with only a few triterpene backbones examined for the CYP716 family members from each plant, biased by the triterpenoids of interest from that particular plant. It is thus perhaps not surprising that activity was found for only approximately a third of the investigated CYP716 family members. However, this does leave in question what contribution the remaining two-thirds (majority) of these CYPs make to triterpenoid biosynthesis. For example, it might then be possible that some of these instead act in other types of natural products biosynthesis (e.g., of sester- or di-terpenoids, much as CYP88 family members have been diverted from gibberellin diterpenoid metabolism to triterpenoid biosynthesis). Accordingly, it seems a bit presumptuous to call the CYP716 family the "cradle of triterpenoid diversity", while it certainly seems to have contributed, the title perhaps overstates the case by some measure. Nevertheless, this manuscript does present an interesting and novel approach to examining the role of CYPs in natural products biosynthesis, and the authors should be applauded for their bold study.*

To avoid making overstatements, we have changed the title of the manuscript to ‘The ancient CYP716 family is a major contributor to the diversification of specialized triterpenoid biosynthesis in eudicots’.

With regard to the CYP716s that are not active in our yeast system, it remains to be determined as to what contribution they may make to triterpenoid biosynthesis. Lack of detected activity for a particular CYP716 family member in our yeast-based assay may have different causes. It is indeed possible that it metabolizes different terpenoid backbones, from sterols over other (tetracyclic) triterpenoids to any other class of terpenoid. For instance, as the CYP716 may only work on a downstream triterpenoid pathway intermediate, thus requiring the prior activity of other P450 enzymes, we may miss detection of triterpenoid metabolizing activity in our current setup. It is noteworthy though that most of the CYP716s for which we could not find a function have been cloned from *A. coerulea* genomic DNA. Hence, it cannot be excluded that some of these genes may not be expressed and thus represent pseudogenes. Further, as we are expressing the plant CYP716 in a yeast system, it cannot be excluded that the enzyme is not being expressed and/or folded in a correct way, and thus may result to be inactive in this heterologous host. Hence, one must be cautious when drawing conclusions on the possible contribution of the ‘non-active’ CYP716s to

triterpenoid biosynthesis. We have incorporated the above reflections in the discussion section of the revised manuscript.

Reviewer #2:

1/ Conceptually, the manuscript by Miettinen et al. addresses the very interesting prospect that cytochrome P450 enzymes mediate the secondary diversification of the amyirin, oleanane, ursane and cycloartane triterpene scaffolds in plants. Although not explicitly stated in the manuscript, the notion seems to be that progenitor P450s with specificity for tetracyclic triterpenes could have given rise to those with pentacyclic specificity, then new alleles neofunctionalized with different regio- and successive oxidation reactions. The authors approach has been to screen 3 plants known for their diversity of oxidized triterpenes for their CYP716 family of CYPs, to functionally characterize these by expression in yeast engineered for amyirin, oleanane, ursane or cycloartenol production, then to chemically profile the yeast for new triterpene products. Finally, the authors appear to use a very common clustering algorithm to suggest some affiliation of CYP716s to particular plant families and lineages.

But does this analysis really establish that the CYP716s are the cradle of chemical diversity of triterpenes?

We have changed the title of the manuscript to ‘The ancient CYP716 family is a major contributor to the diversification of specialized triterpenoid biosynthesis in eudicots’ to avoid making overstatements (see also our reply to the comment of reviewer #1).

2/ There are many experimental components to this work and all are expertly performed. First, they have used a variety of transcriptomic resources to develop databases to screen for the respective CYP716 transcripts. In some cases, they had to generate sequence information themselves and in other cases they needed to generate full-length clones for the desired transcripts identified in the RNA sequence information. They then had to introduce the various CYP genes into yeast producing the various triterpene scaffolds, then profile the yeast extracts by GC-MS, and in at least one case, use LC-MS to identify a temperature labile lactone structure.

There are a few issues the authors may want to consider.

A/ First, the chemical profiles as presented are very aesthetically pleasing, but almost impossible to read. As much as I like these plots, I really want to see more quantitative information and suppose a carefully crafted table could do that, with the chromatograms provided in the supplementary data. I do mean quantitative information because I would like to sum up the total of all the intermediates to gain a pseudo substrate to product ratio. For example, in the control BAS expressing line, 100 units of beta-amyirin. Then in the line expressing BAS plus CYP716A86 or A83, I would expect the total triterpene content to be about the same, 100 units, A83 having much less beta-amyirin relative to A86. If this were not the case, then the authors might have some explaining to do. I'd also like to know that these accumulation patterns are reproducible - 3 replicates within an experiment and certainly with the same results observed in subsequent experiments.

The quantitative analysis is also important when it comes to offering the maps for catalytic steps, as in Figure 2. When the authors observe that CYP enzymes can work in either order of catalysis, it might

be helpful to know if the quantitative yields varied for the initial step. That could be indicative of a preferred order in the cascade.

We thank the referee for this suggestion. We agree that such quantitative information may provide more information on triterpenoid catalysis by the CYP716s. We would like to note however that there is one element in our experimental set-up that complicates quantitative analysis as proposed by the reviewer. We are continuously adapting our yeast platform to improve yield, and along this process, we have started including cyclodextrins in the yeast culture medium (see our PNAS publication by Moses et al., 2014 in the reference list). Cyclodextrins stimulate triterpenoid production in yeast by a not yet fully understood sequestering mechanism that also changes the dynamics of triterpenoid production. As such, contrary to the assumption of the reviewer, the total triterpenoid content does not remain the same in our strains. When a P450 is expressed in combination with an OSC and uses the cyclization product as the substrate, the total triterpenoid content will increase. Likewise, the total triterpenoid content will increase again when multiple active P450s are combined in comparison to strains with only a single active P450. We assume that endogenous accumulation of P450 products imposes negative feedback to the production of the substrate by the OSC, which is removed by sequestration through the cyclodextrins. The use of cyclodextrins has become imperative in our work, however, also for gene discovery programs, since their triterpenoid production boosting effect allows detecting also minor P450 products and greatly enhances the success rate of our functional screens. For the information of the reviewer, the use of cyclodextrins has now been specified clearer in the Methods section (see also our reply to the comment#5 of reviewer #3).

Nonetheless, we could conduct a semi-quantitative analysis by measuring the relative amounts of the triterpenoid products in our yeast strains. We have carried this out for all oleanane-producing yeast strains because they all produced all triterpenoid products in measurable amounts. For each strain four novel biological replicates were sampled, profiled and relative triterpenoid quantities estimated. The triterpenoid compounds were quantified by measuring peak area of extracted ion intensities of representative ions. The semi-quantitative data are represented in the new Table 1 and Supplementary table 3.

As such, our semi-quantitative analysis demonstrated that when a new CYP716 with activity towards a given substrate was introduced in the yeast strain producing that substrate, the levels of the substrate clearly diminished, consolidating that the active CYP716s indeed consume certain substrates in vivo. Likewise, the obtained data also allowed comparing the efficiency of different CYP716 enzymes with similar activity and/or proposing preferred reaction sequences within the biosynthetic pathways. For CYP716s from each of the three plant species investigated an additional paragraph has been included in the respective results section to describe the outcome of the semi-quantitative analysis. Furthermore figures 1d and 2f have been modified to accommodate arrows of different weight reflecting the preferred order of reactions in the biosynthetic pathways. Importantly, production rates of all products was consistent between the four yeast clones, underscoring the reproducibility of the analysis, except for the strains with CYP716E41 from *Centella asiatica*, which showed large quantitative variability although the same products were present in all four replicates.

Finally, we have edited all figures to increase readability (in particular the text font size).

B/ My second concern pertains to the phylogenetic inferences. I'm first assuming the trees are neighbor joining and derived from compiled sequence alignments. Given that the CYP716 family is defined by be greater than 40% identical, the clustering observed could be related to function, but it would be nice if the authors could elaborate on this point. For example, commenting about which members within the trees exhibit different or promiscuous triterpene scaffold preference, followed by regio- and successive oxidation specificities. The cladograms are probable sufficient for inferring general origins, but even that analysis isn't very well described. Given that the authors want the general reader to take away that these CYPs are the cradle of chemical diversity for triterpenes, improving the sophistication and in depth considerations of this part of the work would certainly be welcomed.

The CYP716 phylogenetic analysis in this work has been conducted with the maximum likelihood algorithm of the MEGA5 software package using a sequence alignment produced with the MUSCLE algorithm (see the methods section). Principally, clustering of subgroups may indeed be related to function but this is not straightforward to decipher at this stage. We believe that we do not have enough functional data yet to infer the kind of relationships suggested by the reviewer, with the exception of the CYP716A subgroup. For instance, subgroups S and Y contain several characterized enzymes but most of them have seemingly unrelated triterpenoid backbone preferences and regioselective activities. Subclades E and C in contrast contain only one characterized CYP716 each making it impossible to draw any conclusions.

With regard to promiscuity in triterpenoid backbone preference, all of the subgroups that contain two or more characterized enzymes (i.e. the A, S and Y subgroups), have enzymes that can act on at least two different backbones. This suggests that promiscuity is probably not a specific feature of a single subgroup but rather a demonstration of the flexibility of the CYP716 family.

Nonetheless, more information concerning the origins of different subgroups could indeed be extracted from the cladogram in Fig 4b. We find it particularly intriguing that most of the eudicot CYP716 subgroups were already present in the first eudicots. The corresponding texts in the results and discussion sections have been adapted to incorporate these reflections.

Reviewer #3:

Minor comments:

1/ page 9 line 2: I assume that the authors mean an m/z of 686 and not 586, since it would not make sense.

We thank the reviewer for pointing to a possible error in that sentence. However, the m/z of 586 of the putative hydroxy- β -amyrin is correct but the m/z of 12,13 α -epoxy β -amyrin should be 514 instead of 614. This has been corrected.

2/ yeast-dependent cycloartenol metabolism has been documented on several occasion and this needs to be mentioned (see Gas-Pascual et al PLOS one, 9, (2014)).

The corresponding reference has been included in the revised manuscript.

3/ it is not so clear what the authors mean by "earlier diverged CYP716s", after referring to Amborellales. Do they mean non-existent genes/enzymes?

We apologise for the confusion. The message we were trying to convey was that eudicot species have CYP716s of the "eudicots CYP716s" that have evolved in early eudicots but also CYP716s that have evolved before the divergence of eudicot species but that are nonetheless considered as eudicots and thus have been retained in some eudicot families. We have clarified this by modifying the text to 'Members of the "Angiosperm" subgroups contain CYP716s from several clades such as Magnoliids, Chloranthales and Amborellales but also CYP716s from eudicot species that have likely evolved prior to the divergence of eudicots but are nonetheless considered as eudicots.'

4/ pages 16 and beyond: the two monocot plants showing the presence of CYP716 gene(s) are both Liliales. This could be worth mentioning, if possible in a broader functional or evolutionary context.

As indicated in the text, we consider these genes from *Phormium tenax* (from the Asparagales) and *Xerophyllum asphodeloides* (from the Liliales) to be likely artefacts, perhaps resulting from contamination or a mistake in data handling in the massive sequencing efforts, for instance. Several of such artefacts have already been reported <https://pods.iplantcollaborative.org/wiki/display/iptol/Sample+source+and+purity>. The following facts support our interpretation further: 1) the genome of the earlier diverged monocot *Spirodela polyrhiza* (Alismatales) has been sequenced and it contains no CYP716 sequences, and 2) the 1,000 plant genome project has 66 transcriptomes sequenced from plants diverged prior to the commelinids (from which 8 Liliales and 46 Asparagales species) and no CYP716 sequences could be detected in these sequence data. Hence, we did not discuss possible evolutionary context for these three CYP716 genes further.

5/ it would be worth mentioning, at least in the method section, why MC β D is added to the medium.

The requested information has been added in the Methods section (see also our reply to comment#2A of reviewer#2).

6/ figures: all of them are extremely small and not quite readable. Especially carbon numbering of the formula is not readable at all. Also subfamily names on the right part of figure 4b.

We apologise for this. All figures have been edited to increase readability (in particular the text font size).

Response to the Reviewers

Reviewer #1

This authors have adequately addresses my previously expressed concerns, particularly with the new title and discussion of potential rationales for the many CYP716 family members that do not exhibit activity in the somewhat limited assays that are currently reported.

Reviewer #2

The authors have fully and adequately responded to my previous concerns. I'm especially appreciative of the extra effort given to providing the quantitative information requested in new Table 1. Only one minor suggestion - it would be informative for the general reader if the method used for the phylogenetic tree (maximum likelihood) was stated in the figure legend. The title of the supplementary materials should also be changed to the new title of the manuscript. The new title is actually much better and informative in this reviewer's opinion.

The method information has been added to the legend of the figures with the phylogenetic trees.

Reviewer #3:

I found proper answers to all my minor comments.

I must however express my skepticism concerning the tentative quantification required by reviewer 2. Cytochromes P450, even when closely related, show very different levels of expression in yeast. This is especially true for fast evolving families to which the CYP716 family belongs. In addition, their respective levels of expression in vivo is impossible to quantify. Authors state themselves to their answer to reviewer 1 that they cannot evaluate if a P450 is expressed or not in their whole yeast assay.

Consequently, it is extremely risky to draw quantitative and comparative activity conclusions from experiments carried out using whole yeast assays, as done in this updated work. Personally, I would have refused to compel to Reviewer 2 request for quantitative evaluation, since to my eyes it has very limited chances to be reliable.

We understand and appreciate the concern of the reviewer. Accordingly, we refer in the discussion to the care with which these data should be interpret. Nonetheless, we feel that this quantitative analysis does provide some useful information, in particular with regard to the comparison of substrate consumption by the different CYP716s. For instance, the fact that substrate concentrations go down indicate that these compounds are indeed consumed by the subsequent CYP716s. Likewise, in the case where one CYP716 can use multiple substrates (semi)quantification of products can give us a crude picture about the most important product of the enzyme.